# Biochar Mitigates the Negative Effects of Microplastics on Sugarcane Growth by Altering Soil Nutrients and Microbial Community Structure and Function

**DOI:** 10.3390/plants13010083

**Published:** 2023-12-27

**Authors:** Qihua Wu, Wenling Zhou, Diwen Chen, Jiang Tian, Junhua Ao

**Affiliations:** 1Institute of Nanfan & Seed Industry, Guangdong Academy of Sciences, Guangzhou 510316, China; wqh5859@126.com (Q.W.); zwl2018@vip.163.com (W.Z.); chendiwen@126.com (D.C.); 2Root Biology Center, South China Agricultural University, Guangzhou 510642, China; jtian@scau.edu.cn

**Keywords:** microplastics, biochar, sugarcane biomass, microbial community, red soil

## Abstract

Microplastic pollution in sugarcane areas of China is severe, and reducing the ecological risks is critical. Biochar has been widely used in soil remediation. This study aims to explore the effects and mechanisms of microplastics combined with or without biochar on sugarcane biomass, soil biochemical properties in red soil through a potted experiment. The results show that, compared with control (CK), treatments with microplastics alone reduced the dry biomass of sugarcane, soil pH, and nitrogen (N) and phosphorus (P) contents by an average of 8.8%, 2.1%, 1.1%, and 2.0%, respectively. Interestingly, microplastics combined with biochar could alleviate the negative effects of microplastic accumulation on sugarcane growth and soil quality. There were significant differences in the bacterial community alpha diversity indices and compositions among different treatments. Compared with CK, treatments with microplastics alone obviously decreased the observed operational taxonomic units (OTUs) and the Chao1 and Shannon indices of soil total bacteria (16S rRNA gene-based bacteria) while increasing them in *phoD-harboring* bacteria. Microplastics combined with biochar treatments significantly increased the abundance of *Subgroup_10* for the 16S rRNA gene and treatments with microplastics alone significantly increased the relative abundance of *Streptomyces* for the *phoD* gene compared to CK. Moreover, compared with microplastics alone, the treatments with microplastics combined with biochar increased the relative abundance of *Subgroup_10*, *Bacillus*, *Pseudomonas* in soil total bacteria, and *Amycolatopsis* and *Bradyrhizobium* in *phoD-harboring* bacteria, most of which can inhibit harmful bacteria and promote plant growth. Additionally, different treatments also changed the abundance of potential microbial functional genes. Compared to CK, other treatments increased the abundance of aerobic ammonia oxidation and denitrification but decreased the abundance of nitrate respiration and nitrogen respiration; meanwhile, these four functional genes involved in N cycling processes were obviously higher in treatments with microplastics combined with biochar than in treatments with microplastics alone. In conclusion, microplastics combined with biochar could alleviate the negative effects of microplastic accumulation on sugarcane biomass by altering soil nutrients and microbial community structure and function.

## 1. Introduction

The prevalence of microplastic (plastic particles < 5 mm) pollution in the world and its potential animal, plant, and human health risks have attracted great attention in recent years [1,2]. Terrestrial ecosystems are an important gathering place of microplastics, so the impact of microplastics on soil ecosystems and plant growth and development has become a research hotspot [3]. The possible sources of microplastics in the soil environment mainly include plastic film mulching, sludge landfill, compost application, and irrigation [4]. Plastic film mulching is an effective measure used to increase crop yield. Sugarcane mulching has been applied in China for more than 30 years, and there is basically no recovery of plastic film in sugarcane fields [5]. Our investigation in the sugarcane area showed that the amount of plastic film used in sugarcane fields was 75–150 kg/ha/year. Assuming that all the plastic film is converted into microplastics and remains in the soil, the cumulative concentration of microplastics in the soil of the sugarcane area would reach 0.1–0.2% in 30 years, which indicates that the pollution of microplastics in sugarcane fields is serious. At the same time, sugarcane has the characteristics of a long growth cycle, high yield, and high nutrient demand [5,6]. Therefore, it is particularly necessary to explore the effects of microplastic pollution on sugarcane growth and soil nutrient content.

When microplastic particles enter the soil, they can combine with soil particles, thereby affecting the physical and chemical properties of the soil, the distribution of soil microorganisms, and the growth and development of plants [7,8]. Several researchers have found that microplastics affect soil physical and chemical properties, including soil porosity and moisture, pH, organic matter, and N, P, and potassium (K) nutrient content [4,9]. Polyethylene (PE) microplastics have been found to have increased soil pH and the content of soluble organic matter and ammonium N by microcosm incubation [10]. Microplastics have also been reported by many studies to affect soil microbial community structure and function [11,12]. Microplastics can directly or indirectly change soil properties, thereby affecting the composition and diversity of microbial communities [13,14]. At the same time, microplastics themselves can provide specific substrates or adsorption sites for microorganisms [15]. The microbial community structure in the biofilm of microplastics is clearly different from the microbial composition in the soil, thus changing the functional properties of the soil [12]. In addition, microplastics can also cause changes in rhizosphere microbial communities, which may affect plant growth and development to a certain extent [9]. However, few studies have focused on the effects of microplastic accumulation on soil microorganisms, especially those related to P cycling, and the relationship between microbial changes and sugarcane growth.

Biochar is formed by the pyrolysis of biomass under anaerobic conditions. Its carbon (C) content can reach 40–75%, and it is rich in mineral nutrients, functional groups, and pore structure. Biochar is stable and not easy to degrade, so it has great application prospects in promoting crop growth, soil improvement, and soil pollution remediation [16]. Biochar can alleviate the impact of microplastic pollution on soil microbial diversity [17,18]. It has been found that polyvinyl chloride (PVC) microplastic pollution has adverse effects on crop yield, soil enzyme activity, and microorganisms, whereas biochar mitigates the adverse effects [18]. However, to date, the effects and mechanisms of biochar application on sugarcane growth, soil nutrients, and microbial communities in soils contaminated with different concentrations of microplastics are still unclear.

In this study, sugarcane planting soil was collected to explore the effects of microplastics combined with or without biochar on sugarcane biomass soil biochemical properties through pot experiments. The study aimed to (1) identify the differences in the biomass of sugarcane soil physiochemical properties and microbial community characteristics under different treatments of PE plastics combined with or without corn straw biochar and (2) explore the relationship among sugarcane biomass soil physiochemical properties and microbial community characteristics and their mechanism. Based on previous research results [19] and sugarcane field survey data, we hypothesized that PE microplastics added alone will decrease the biomass of sugarcane and have a negative impact on soil physicochemical and microbial communities, whereas the addition of biochar may alleviate the negative effects of soil microplastic pollution on plant growth and microbial community structure and function, as biochar has rich nutrients and a large specific surface area. The purpose of this study is to provide a reference for the control of microplastic pollution in sugarcane fields and promote the green development of the sugarcane industry.

## 2. Materials and Methods

### 2.1. Experimental Soil, Biochar, and Microplastics

The tested red soil was collected from the sugarcane experimental base in Wengyuan, Guangdong (113.94° E, 24.28° N), and the soil physicochemical properties were as follows: pH 4.14, organic matter 14.2 g/kg, available N 158.67 mg/kg, available P 17.47 mg/kg, and available K 89.50 mg/kg. Corn stalk biochar (BC) was purchased from Henan Lize Environmental Protection Technology Co., Ltd. (Zhengzhou, China), and was obtained by pyrolysis at 600 °C. Its main properties were pH 9.0, organic carbon 51.1%, total N 0.9%, total phosphorus 0.2%, and total potassium 1.6%. High-density polyethylene microplastics (PEs) were purchased from Shanghai Aladdin Company. According to the data provided by the manufacturer, the polyethylene particles have a random spherical structure, a purity of 99.99%, a density of 0.96 g/cm^3^, and a melting point of 132°. In this study, polyethylene spherical particles with a particle size of approximately 180 µm were selected for the experiment. PE microplastics were first placed in a solution to remove possible heavy metals, then washed with deionized water and air-dried, sterilized to eliminate microbial contamination, and stored in a refrigerator at 4 °C for later use.

### 2.2. Experimental Setup and Design

The pot experiment was conducted in a greenhouse, and six treatments were set up: adding 0.1% PE microplastic (*w*/*w*, soil dry weight, low PE), adding 1% PE microplastic (high PE), adding 1% BC and 0.1% PE microplastic combined with 1% biochar (low PE + BC), 1% PE microplastic combined with 1% biochar (high PE + BC), and control (no addition of PE microplastic or biochar, CK). Each treatment was repeated three times, with a total of 18 pots arranged randomly. The pot was 30 cm high, with a top diameter of 32 cm and a bottom diameter of 26 cm. Urea was used as the main source of N at a rate of 300 mg N/kg soil, and potassium dihydrogen phosphate was used as the source of P and K at a rate of 150 mg K/kg soil. Soil micronutrient additions were (mg/kg soil) CaCl_2_ 125.67, MgSO_4_·7H_2_O 43.34, EDTA-FeNa 5.80, and MnSO_4_·4H_2_O 6.67. Before potting, all of the above microplastics, biochar, and nutrients were fully mixed with the soil. On 1 March 2022, one healthy sugarcane seedling (ROC22) of similar growth was transplanted into each pot. During the experiment, the weighing method was used to keep the water content at 70–80% of the field capacity to meet the water demand of sugarcane. It was harvested on 30 December 2022.

### 2.3. Sample Collection and Analysis

#### 2.3.1. Plant and Soil Collection

Plant samples were collected, including aboveground (cane and leaves) and underground (root) parts. After cleaning, the sugarcane plants were quenched at 105 °C for 30 min and dried at 70 °C to constant weight, and the dry biomass was weighed [20]. For the collection and treatment of soil samples, the rhizosphere soil was obtained by shaking the soil, and the test plants were gently shaken to remove the larger soil clumps from the root system. Then, the soil attached to the root system was shaken off and put into a sterile self-sealing bag. Soil samples were quickly brought back to the laboratory, and one part of the fresh sample of rhizosphere soil was stored in a refrigerator at −40 °C for analysis of soil microorganisms. The other part of the rhizosphere soil was naturally air-dried and then used to determine the physicochemical properties of the soil after impurities were removed.

#### 2.3.2. Measurement of Soil Physicochemical Properties

Soil pH was measured with a pH meter (1:2.5 soil/water). The electrical conductivity (EC) was determined by an electrical conductivity meter. Soil organic carbon (SOC) was measured by the digestion method. Soil total N and total K were determined using the micro-Kjeldahl digestion and flame photometry methods, respectively. Soil available N was determined using the alkaline diffusion method. Available K was extracted with 1 M NH_4_OAc and then measured with a flame photometer. Total P was extracted with an acid solution (H_2_SO_4_-HClO_4_), and the available P was extracted with 0.5 NaHCO_3_ (Olsen P) and then measured through molybdenum blue colorimetry [21].

#### 2.3.3. Soil DNA Extraction and Microbial Community Analysis

Soil DNA was extracted from 0.5 g of fresh soil samples using a Power Soil DNA isolation kit (MoBio Laboratories, Carlsbad, CA, USA) following the instructions in the accompanying manual, and each treatment contained 3 biological replicates. The quality and concentration of the extracted DNA were determined with a Qubit™ 3.0 fluorometer (Thermo Fisher Scientific Inc., Waltham, MA, USA). The extracted DNA samples were stored at −40 °C.

The 16S rRNA gene was amplified with the primer F515/R907 (GTGCCAGCMGCCGCGG/CCGTCAATTCMTTTRAGTTT) using the qualified DNA of the assay sample as the template. PCR was carried out on a MasterCycler gradient (Eppendorf, Germany) using 25 μL reaction volumes containing 5 μL 5 × reaction buffer, 5 μL 5 × GC buffer, 2 μL dNTP (2.5 mM), 1 μL forward primer (10 µM), 1 μL reverse primer (10 µM), 2 μL DNA template, 8.75 μL ddH_2_O, and 0.25 μL Q5 DNA polymerase. The cycling parameters were 95 °C for 5 min, followed by 28 cycles of 95 °C for 45 s, 55 °C for 50 s, and 72 °C for 45 s, with a final extension at 72 °C for 10 min. Each sample was mixed after three replicates at the same time. Agarose gel electrophoresis (1%) was used to detect the quality of PCR amplification products, an Omega gel recovery and purification kit was used to purify them, and Qubit 2.0 was used to determine the concentration of purified PCR products. The PCR amplification products of different samples were mixed based on equimolar amounts, and the quality of the mixed PCR products was detected at the same time. Then, sequencing was completed by the Illumina MiSeq-PE300 sequencing platform.

In addition, primers ALPS-F730/ALPS-R1101 (CAGTGGGACGACCACGAGGT/GAGGCCGATCGGCATGTCG) were used for *phoD* gene amplification [22]. PCR was carried out on a Mastercycler gradient thermal cycler (Eppendorf, Germany) using 25 μL reaction volumes containing 5 μL 5 × reaction buffer, 5 μL 5 × GC buffer, 2 μL dNTP (2.5 mM), 1 μL forward primer (10 µM), 1 μL reverse primer (10 µM), 2 μL DNA template, 8.75 μL ddH_2_O, and 0.25 μL Q5 DNA polymerase. The cycling parameters were 94 °C for 5 min, followed by 35 cycles of 94 °C for 30 s, 50 °C for 30 s, and 72 °C for 60 s, with a final extension at 72 °C for 7 min. Sequencing was completed by the Illumina MiSeq-PE250 sequencing platform after they were qualified.

The raw data were first screened and qualified and then separated using barcode sequences and trimmed with Trimmomatic (version 0.36) after the PCRs were carried out. Then, the dataset was analyzed using QIIME (v1.9). The sequences were clustered into OTUs by VSEARCH (v2.13.4) at a similarity level of 97%. The Basic Local Alignment Search Tool (BLAST) (v2.6.0) was used to classify all the sequences into taxonomic groups based on the Silva database. Then, alpha diversity indices (observed OTUs, Chao1, and Shannon indices) were calculated with Mothur software (version 1.30.1).

### 2.4. Data Analysis

All data were collated using Excel; SASV9 software was used for one-way analysis of variance, and the Duncan (SSR) method was used to test the significance of the mean value (3 replicates) of each index among different treatments (*p* < 0.05). The data of the species were analyzed by detrended correspondence analysis (DCA), and the length of the first DCA axis was 1.4 for 16S rRNA gene-based bacteria and 3.3 for *phoD*-*harboring* bacteria. Then, the relationships between soil microbial community structures and soil environmental factors were analyzed by redundancy analysis (RDA). Spearman’s correlation coefficients were employed to test the relationships between the soil bacteria, functional genes, and soil properties.

## 3. Results

### 3.1. Biomass of Sugarcane

The aboveground biomass (AGB, the sum of cane and leaves) and below-ground biomass (BGB, the root) of sugarcane in different treatments showed some differences (Figure 1). Compared with the CK treatment, the AGB of the BC treatment significantly increased by 7.1%; the AGB and BGB of the low PE and high PE treatments significantly decreased by 4.2% and 8.6% and 22.6% and 37.9%, respectively; and the BGB of the low PE + BC and high PE + BC treatments significantly decreased by 8.8% and 21.9% (*p* < 0.05), respectively. In addition, the total biomass of sugarcane (the sum of BGB and BGB) was in the order of BC > low PE + BC > CK > high PE + BC > low PE > high PE. The AGB and BGB in the treatments with microplastics combined with biochar (low PE + BC and high PE + BC) were significantly higher than those in the treatments with microplastics alone (low PE and high PE).

### 3.2. Physical and Chemical Properties of Soil

Compared with CK, treatments with microplastics alone decreased the pH values averagelly by 2.1%, whereas treatments with BC amendment of low PE + BC and high PE + BC and BC increased the pH values by 2.5%, 1.0% and 3.0%, respectively (Table 1). The other five treatments all increased the SOC contents compare to CK, but only treatments with BC amendment were significantly higher than CK (*p* < 0.05). Compared with CK, the contents of total N, P, and K and available N and P decreased on average by 1.1%, 2.0%, 0.6%, 1.2%, and 13.5%, respectively, whereas the available K content increased by 1.8% in treatments with microplastics alone; the corresponding indices in treatments with BC amendment significantly increased. In addition, the contents of pH, EC, SOC, total N, total K, and available N, P, and K in the treatments of microplastics combined with biochar were significantly higher than those in treatments with microplastics alone.

### 3.3. Soil Microorganisms

#### 3.3.1. Diversity of Soil Microbial Communities

For the total bacteria (16S rRNA gene-based bacteria), compared with CK, the observed_OTUs and the Chao1 and Shannon indices of low PE and high PE were decreased by 5.0%, 5.1%, and 1.4% and by 7.5%, 7.3%, and 2.1%, respectively, whereas these three indices were increased for low PE + BC, high PE + BC, and BC treatments (Table 2). Moreover, compared with the treatment with microplastics alone, the treatments with microplastics combined with biochar significantly increased the observed_OTUs and the Chao1 and Shannon indices. For the *phoD*-*harboring* bacteria, compared with CK, the observed_OTUs and the Chao1 and Shannon indices of the other five treatments were significantly increased, and these indices were the highest in the high PE and lowest in the low PE treatments. In addition, the treatments of microplastics combined with biochar reduced the observed OTUs and the Chao1 and Shannon indices compared with the treatment with microplastics alone.

#### 3.3.2. Soil Microbial Community Composition

There were significant differences in the composition and proportion of dominant bacterial species in soil samples from different treatments (Figure 2). For total bacteria (16S rRNA gene-based bacteria), at the phylum level, the dominant bacteria were Proteobacteria, Chloroflexi, Actinobacteria, Acidobacteria, and Gemmatimonadota in all treatments, with average abundances of 41.0%, 9.1%, 11.8%, 9.0%, and 8.0%, respectively. There were significant differences in the relative abundances of two of the top ten bacterial phyla (*p* < 0.05) among the six treatments (Appendix A). Compared with CK, the other five treatments significantly decreased the relative abundance of Gemmatimonadota, and increased abundance of Acidobacteria was observed in the BC treatment. In addition, the treatments of microplastics combined with biochar increased the relative abundance of Acidobacteria compared with the treatments adding microplastics alone. At the genus level, the top five genera were *Pedosphaeraceae*, *Dongia*, *TRA3-20*, *Sphingomonas*, and *Subgroup_10* among all treatments, and the average abundances were 2.1%, 2.4%, 3.0%, 2.8%, and 2.1%, respectively. There were significant differences in the relative abundances of seven of the top ten bacterial genera (*p* < 0.05) among the six treatments (Appendix A). However, for the top 10 genera, there was no significant difference between low PE or high PE with CK, only the treatments of low PE + BC and high PE + BC significantly increased the relative abundance of *Subgroup_10* but significantly decreased abundance of *TRA3-20*, *SC-I-84*, and *Ellin6067* compared to CK. In addition, treatments of microplastics combined with biochar increased the relative abundance of *Subgroup_10*, *Bacillus*, and *Pseudomonas* but decreased the abundance of *TRA3-20*, *SC-I-84*, and *IMCC26256* compared with the treatments with microplastics alone.

For *phoD*-*harboring* bacteria, the dominant phyla in all treatments were Proteobacteria and Actinobacteria at the phylum level, and the sum of their relative abundance was in the range of 89.9–94.6%. There was no significant difference in the relative abundance of Proteobacteria and Actinobacteria among other five treatments with CK (Appendix A). At the genus level, *Amycolatopsis*, *Streptomyces*, and *Bradyrhizobium* were abundant in all treatments (at least one treatment > 1%). There was no significant difference in the relative abundances of *Amycolatopsis* or *Bradyrhizobium* among the other five treatments with CK, and only the low PE and high PE treatments significantly increased the relative abundance of *Streptomyces* compared with CK (Appendix A). In addition, the treatments with microplastics combined with biochar increased the relative abundance of *Amycolatopsis* and *Bradyrhizobium* but decreased the abundance of *Streptomyces* compared with the treatments with microplastics alone.

#### 3.3.3. Soil Microbial Function

FAPROTAX was used to predict the potential functions of total bacterial communities. In general, bacterial functions were mainly related to the cycling of C and N in the ecosystem, and there were obvious differences in the abundance of functional genes in different treatments (Figure 3). Compared to CK, the other five treatments increased the abundance of functional genes such as aerobic ammonia oxidation, denitrification, and iron respiration but decreased the aromatic compound degradation, methylotrophy, nitrate respiration, and nitrogen respiration. Moreover, high PE had a significantly higher abundance of functional genes of aerobic chemoheterotrophy, chemoheterotrophy, iron respiration, and photoheterotrophy but had a significantly lower abundance of functional genes of aromatic hydrocarbon degradation and chitinolysis than CK. In addition, compared to the treatments with microplastics alone, treatments with microplastics combined with biochar increased the abundance of aerobic ammonia oxidation, denitrification, nitrate respiration, and nitrogen respiration.

### 3.4. Correlations between Sugarcane Biomass, Soil Microbes, and Soil Physicochemical Properties

RDA was used to analyze the relationship between bacterial community structure and soil properties (Figure 4). For total bacteria (16S rRNA gene-based bacteria), pH, EC, SOC, TN, TP, TK, AN, AP, and AK together accounted for 53.1% of the bacterial community structure variation, with the first two axes accounting for 27.4% and 13.0% of the variation, respectively. The total bacterial (16S rRNA gene-based bacteria) community structure was mainly affected by pH, EC, and AK and was significantly negatively correlated with pH (*p* < 0.05). For *phoD*-*harboring* bacteria, the main physicochemical properties of the soil explained 40.3% of the bacterial community changes, and the first two axes explained 27.5% and 9.8% of the changes, respectively. The community structure of *phoD*-*harboring* bacteria was significantly negatively correlated with SOC, EC, and AK (*p* < 0.05).

The correlation between sugarcane biomass, soil bacterial community composition, functional gene abundance, and soil physical and chemical properties was further analyzed (Figure 5). For total bacteria (16S rRNA gene-based bacteria), the relative abundance of *Subgroup_10* was significantly and positively correlated with sugarcane AGB and BGB, soil pH, EC, SOC, and N, P, and K nutrient contents; the relative abundance of *IMCC26256* was significantly negatively correlated with the AGB and BGB of sugarcane and soil TP; and the relative abundance of *TRA3-20* was significantly negatively correlated with soil SOC, EC, TN, TK, AN, and AK (*p* < 0.05). For *phoD*-*harboring* bacteria, *Streptomyces* had a significant negative correlation with the AGB and BGB of sugarcane, soil pH, TN, AN, and AP, whereas *Bradyrhizobium* had a significant positive correlation with the AGB of sugarcane and soil AP. There was a significant positive correlation between *Saccharopolyspora* and soil TP (*p* < 0.05). For the abundance of functional genes, nitrate respiration and nitrogen respiration were positively correlated with soil pH and AP, whereas fermentation was negatively correlated with soil pH and AP. Denitrification and nitrite respiration were positively correlated with soil AN and EC, whereas methanol oxidation and methylotrophy were negatively correlated with AP. Aerobic ammonia oxidation was negatively correlated with AK and TK.

## 4. Discussion

### 4.1. Effects of Microplastics and Biochar on Sugarcane Biomass and Soil Physical and Chemical Properties

In this study, the addition of microplastics alone significantly decreased the AGB and BGB of sugarcane compared with the CK, and the decreasing amplitude in the high PE treatment was greater (Figure 1), indicating that the accumulation of microplastics would inhibit the biomass of sugarcane and that the degree of influence increased with the increasing concentration of microplastics. This is similar to the findings of many previous studies that the accumulation of microplastics can inhibit plant growth and development, thereby reducing plant [8] biomass [8,9]. Compared with the treatments with microplastics alone, the combination of microplastics with biochar significantly increased the total biomass of sugarcane, which confirmed that the addition of biochar could reduce the inhibition of microplastic accumulation on plant biomass [17,23]. Biochar addition has been found to be able to increase shoot dry matter production in different concentrations of PVC-contaminated soil [18]. At the same time, treatments with microplastics alone altered soil physicochemical properties, including the pH and total and available N and P contents (Table 1). Indeed, MPs like high-density PE have been proven to decrease soil pH through cation/proton exchange [24]. Microplastics have a high adhesion that can change plant root exudates and soil buffering capacity to co-mediate soil pH [9]. Soil microbial community composition is an important indicator of soil quality, and microbial activities also affect the recycling of N, P, and K nutrients [25]. Microplastic addition considerably changed microbial community and activities, which may be an important reason for the alteration in soil N and P content [26]. Many previous studies have reported the effects of PE microplastics on soil physical and chemical properties, but the results varied [19]. For example, the addition of PE microplastics reduced soil pH and organic matter content in red soil [27]. The discrepancy in the effects of PE microplastics on soil properties may be related to the concentration of microplastics, soil type, and climate [4]conditions [4,19]. Additionally, treatments with BC amendment had significantly higher soil pH, SOC, and nutrients than treatments with microplastics alone, which may have been due to the high pH, SOC, and nutrients contents of corn straw biochar itself [28]. In addition, the correlation analysis showed that the total biomass of sugarcane was significantly positively correlated with soil pH, SOC, and total and available N and P contents, indicating that the promotion of sugarcane biomass via biochar amendment may have been due to its improvement to soil quality [29].

### 4.2. Effects of Microplastics and Biochar on Bacterial Community Diversity and Composition

The diversity and composition of total bacteria (16S rRNA gene-based bacteria) were significantly affected by the addition of microplastics or biochar. Previous studies have also reported that microplastic accumulation or biochar amendment changed the diversity of bacteria [13,28]. The treatments with microplastics alone decreased the observed OTUs and the Chao1 and Shannon indices, whereas microplastics combined with biochar increased the three indices compared to CK (Table 2). From the perspective of the abundance and diversity of soil microbial communities, biochar addition can increase the abundance and diversity of soil total bacteria, thereby enhancing the stability and functional diversity of soil ecosystems [30,31]. For total bacteria, the other five treatments significantly decreased the relative abundance of Gemmatimonadota compared with CK (Figure 2 and Appendix A), which may have been due to the inhibition of microplastics accumulation on the underground growth of sugarcane. Several bacteria in Gemmatimonadota form symbiotic relationships with plant roots [32], and the underground biomass of sugarcane treated with microplastics in this study was lower than that of CK, resulting in a decrease in the relative abundance of Gemmatimonadota. The BC treatment significantly increased the relative abundance of Acidobacteria compared with CK, which may have been related to the obvious increase in soil pH, SOC, and nutrients in BC [33]. At the genus level, compared with CK, only treatments with microplastics combined with biochar significantly increased the relative abundance of *Subgroup_10* and significantly decreased the abundance of *TRA3-20*. Correlation analysis showed that the relative abundance of *Subgroup_10* was significantly positively correlated with the AGB and BGB of sugarcane and the main soil physicochemical properties, whereas the relative abundance of *TRA3-20* was significantly negatively correlated with the majority of soil physicochemical properties. Additionally, compared with the treatment with microplastics alone, the treatment with microplastics combined with biochar increased the relative abundance of *Bacillus* and *Pseudomonas* but decreased the abundance of *IMCC26256*. *Bacillus* and *Pseudomonas* are important bacterial groups that can inhibit harmful bacteria and pathogens and promote plant growth. Therefore, the promoting effect of microplastics combined with biochar on sugarcane biomass may be related to changes in the total bacterial community composition in soil [34].

The diversity and composition of *phoD*-*harboring* bacteria in different treatments were also significantly different. The *phoD* gene of bacteria encodes alkaline phosphatase, which plays an important role in soil organic P decomposition [35]. Compared with CK, the observed OTUs and the Chao1 and Shannon indices of the other five treatments were significantly increased, and the index of the high PE treatment was the highest. In this study, the AP in the high PE treatment was the lowest, whereas the abundance and diversity of *phoD*-*harboring* bacteria were the highest, which may be explained by sugarcane requiring a certain amount of P provided by organic P decomposition for growth [36]. For *phoD*-*harboring* bacteria community composition, Proteobacteria and Actinobacteria were the two dominant phyla, both of which contained many currently known P solubilizing bacteria [37]. At the genus level, only the low PE and high PE treatments significantly increased the relative abundance of *Streptomyces* compared with CK (Figure 2 and Appendix A). The higher relative abundance of *Streptomyces* at higher concentrations of microplastics (such as high PE) may have been due to its greater stress tolerance [38]. Moreover, compared with the treatment with microplastics alone, the treatment with microplastics combined with biochar increased the relative abundance of *Amycolatopsis* and *Bradyrhizobium*, both of which are plant growth-promoting bacteria [39]. Correlation analysis also confirmed that *Bradyrhizobium* was significantly positively correlated with sugarcane biomass and soil AP content, whereas *Streptomyces* was significantly negatively correlated with sugarcane biomass, soil pH, TN, AN, and AP (*p* < 0.05). These results confirm that biochar amendment could change the microbial composition and thus alleviate the inhibitory effect of microplastics on plant growth [40].

### 4.3. Effects of Microplastics and Biochar on Microbial Community Structure and Function

Changes in soil physical and chemical properties can cause changes in the soil microbial community structure and further change the function of soil microorganisms [41,42]. RDA showed that the community structure of total bacteria was mainly affected by pH, EC, TN, and AK, and the structure of the total bacterial community was significantly negatively correlated with pH (*p* < 0.05). Soil pH, EC, TN, and AK increased significantly with biochar addition, which in turn drove significant changes in the total bacterial community structure (Table 1 and Figure 4). Other studies also showed that the soil total bacterial community structure was affected by soil pH, SOC, AP, and other physical and chemical properties [37,43]. Soil pH and NO_3_^−^ concentration played an important role in the formation of bacterial community structure in black soil [43], whereas the bacterial community structure on loess was significantly correlated with dissolved organic C, AP, and TP [37]. These results confirm that there were some differences in the factors affecting the structure of the total bacterial community in soil, which may have been due to differences in the soil types and fertilization management measures. The structure of *phoD*-*harboring* bacteria community was significantly negatively correlated with SOC, EC, and AK (*p* < 0.05). The results of different studies on the relationship between the structure of *phoD*-*harboring* bacteria community and soil physicochemical properties are quite [35] different [35,36]. Chen et al. [36] reported that soil pH and AP were related to changes in the community structure of *phoD*-*harboring* bacteria in soil, whereas Liu et al. [37] found that various physical and chemical properties had little effect on the structure of *phoD*-*harboring* bacteria community. These variable findings may imply that the structure of the *phoD*-*harboring* bacteria community is quite different under different soil types and climatic conditions [37].

Correlation analysis showed that AP and pH significantly affected soil microbial species and the abundance of functional genes (Figure 5). Soil pH was significantly correlated with *Subgroup_10* and *Streptomyces*, as well as nitrate respiration, nitrogen respiration, and fermentation. Similarly, found that pH was significantly correlated with nitrogen respiration and nitrate respiration [41]. Furthermore, there were significant differences in the abundance of functional genes among the different treatments. Compared with CK, the high PE treatments significantly changed the abundance of functional genes related to C cycling such as aerobic chemoheterotrophy, chemoheterotrophy, aromatic hydrocarbon degradation, and chitinolysis (Figure 3). The high functional genes related to C cycling in the high PE treatment may have been related to the changes in the content and form of C in the soil after adding a high concentration of microplastics, for microplastics are high-C polymers with a C content of more than 90% [44]. Moreover, compared with the treatment with microplastics alone, the treatment with microplastics combined with biochar increased the abundance of functional genes such as aerobic ammonia oxidation, denitrification, nitrate respiration, and nitrogen respiration. All of these functional genes participate in the soil N cycle and may affect the soil N content available to the plant [41]. The increase in soil AN content and the biomass of sugarcane in treatments with microplastics combined with biochar may also have been related to the alteration to functional genes after the biochar amendment [17].

## 5. Conclusions

The accumulation of polyethylene microplastics decreased both the aboveground biomass and underground biomass of sugarcane, soil pH, and nitrogen and phosphorus content, and the degree of influence increased with the increasing concentration of microplastics. In contrast, microplastics combined with biochar could alleviate the negative effects of microplastic accumulation on sugarcane biomass and soil quality. There were significant differences in the bacterial community alpha diversity indices and compositions among different treatments. Treatments with microplastics alone decreased the observed OTUs and the Chao1 and Shannon indices of soil total bacteria (16s rRNA gene-based bacteria) while increasing the three indices in *phoD*-*harboring* bacteria compared to the control. Compared with microplastics alone, the treatments with microplastics combined with biochar increased the relative abundance of *Subgroup_10, Bacillus,* and *Pseudomonas* in soil total bacteria and *Amycolatopsis* and *Bradyrhizobium* in *phoD*-harboring bacteria, most of which can inhibit harmful bacteria and promote plant growth. In addition, compared with the treatments with microplastics alone, the treatments with biochar amendment increased the abundance of functional genes involved in the nitrogen cycle, such as aerobic ammonia oxidation, denitrification, nitrate respiration, and nitrogen respiration, which may have increased the soil nitrogen content available to the plant, thereby promoting the biomass of sugarcane. Overall, our results prove that biochar amendment may alleviate the negative effects of microplastic accumulation on sugarcane biomass by altering soil nutrients and microbial community structure and function.

## Figures and Tables

**Figure 1 plants-13-00083-f001:**
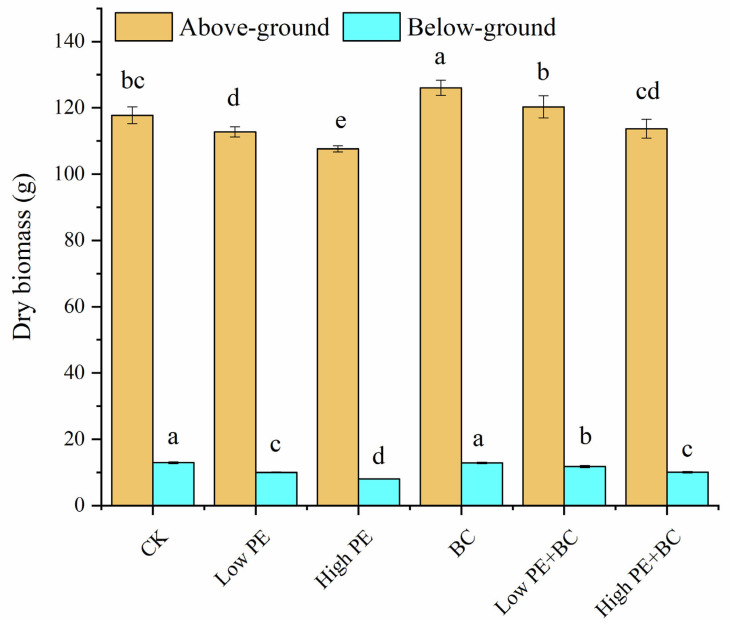
Biomass of sugarcane under different treatments. Different lowercase letters indicate significant difference among the six treatments (*p* < 0.05).

**Figure 2 plants-13-00083-f002:**
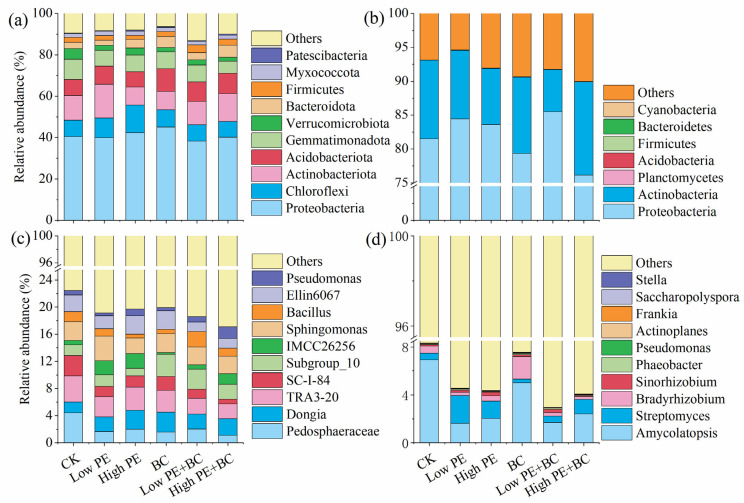
Community composition of total bacteria (top 10 phyla (**a**) and genera (**c**) in abundance) and *phoD*-*harboring* bacteria (phyla (**b**) and top 10 genera (**d**) in abundance) in soils under different treatments.

**Figure 3 plants-13-00083-f003:**
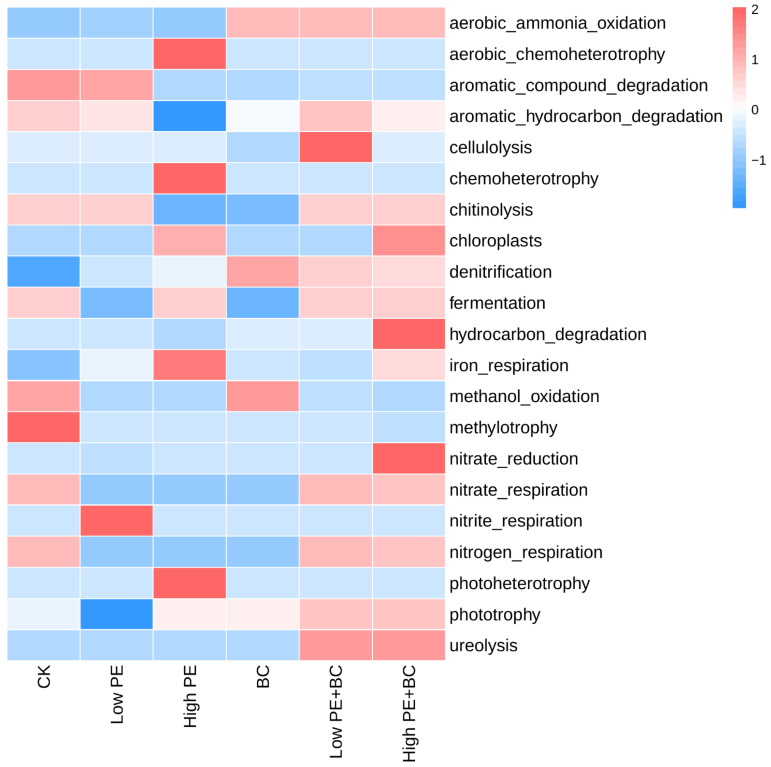
Heatmap of soil functional gene abundance in different treatments.

**Figure 4 plants-13-00083-f004:**
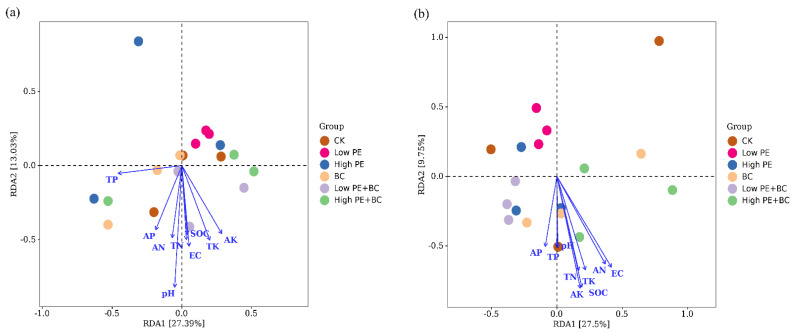
Redundancy analysis (RDA) of soil total bacteria (**a**), *phoD*-harboring bacteria (**b**), and soil physical and chemical properties.

**Figure 5 plants-13-00083-f005:**
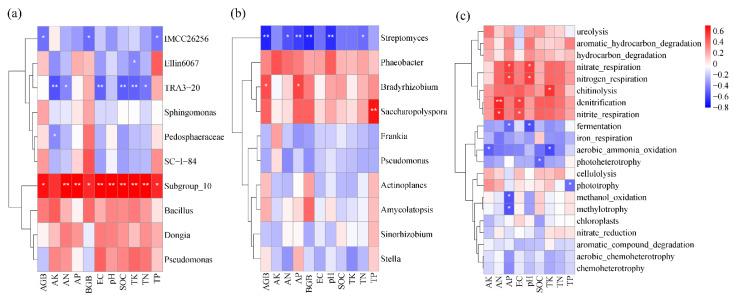
Correlation between soil total bacterial genera (**a**), *phoD*-*harboring* bacteria genera (**b**), ecosystem function (**c**), and soil physical and chemical properties. *, **, significant at *p* ≤ 0.05, *p* ≤ 0.01, respectively.

**Table 1 plants-13-00083-t001:** Changes in physical and chemical properties of soil under different treatments.

	pH (H_2_O)	EC (ms/cm)	SOC (g/kg)	TN (g/kg)	TP (g/kg)	TK (g/kg)	AN (mg/kg)	AP (mg/kg)	AK (mg/kg)
CK	3.96 (0.04) bc	0.82 (0.01) b	12.71 (0.69) b	1.46 (0.03) b	0.54 (0.01) b	14.34 (0.03) c	196.02 (4.06) b	15.55 (2.79) cd	84.67 (3.06) b
Low PE	3.89 (0.04) cd	0.81 (0.02) b	13.32 (0.73) b	1.45 (0.02) b	0.53 (0.02) b	14.31 (0.20) c	194.73 (7.07) b	14.08 (1.41) cd	86.33 (3.51) b
High PE	3.86 (0.05) d	0.83 (0.02) b	13.37 (0.53) b	1.44 (0.03) b	0.53 (0.03) b	14.21 (0.12) c	193.05 (5.41) b	12.84 (0.33) d	86.01 (4.00) b
BC	4.08 (0.04) a	1.06 (0.07) a	19.14 (1.16) a	1.66 (0.02) a	0.60 (0.04) a	15.05 (0.32) b	215.69 (5.81) a	24.44 (1.46) a	95.33 (4.51) a
Low PE + BC	4.06 (0.04) a	1.07 (0.04) a	19.83 (0.49) a	1.66 (0.02) a	0.56 (0.02) ab	15.45 (0.10) a	213.92 (4.17) a	19.31 (1.66) b	97.67 (7.09) a
High PE + BC	4.00 (0.06) ab	1.11 (0.01) a	19.65 (0.37) a	1.66 (0.05) a	0.55 (0.04) ab	15.41 (0.15) a	211.63 (1.6) a	16.38 (1.99) bc	98.67 (4.51) a

Different lowercase letters in the same column indicate significant difference among the treatments (*p* < 0.05).

**Table 2 plants-13-00083-t002:** Bacterial diversity index of soils under different treatments.

	Treatment	Observed OTUs	Chao1	Shannon
16S rRNA gene-based bacteria	CK	1539 (67) ab	1541 (66) ab	9.34 (0.07) b
	Low PE	1462 (73) bc	1463 (37) bc	9.21 (0.05) c
	High PE	1424 (48) c	1428 (48) c	9.14 (0.03) c
	BC	1590 (31) a	1626 (56) a	9.63 (0.08) a
	Low PE + BC	1554 (43) ab	1557 (42) ab	9.56 (0.01) a
	High PE + BC	1538 (66) ab	1542 (67) ab	9.55 (0.03) a
*phoD*-*harboring* bacteria	CK	934 (68) c	1282 (36) c	3.59 (0.18) c
	Low PE	1207 (69) b	1504 (82) b	4.58 (0.20) ab
	High PE	1361 (47) a	1749 (75) a	4.84 (0.24) a
	BC	1197 (39) b	1557 (73) b	4.63 (0.09) ab
	Low PE + BC	1179 (44) b	1465 (38) b	4.51 (0.11) b
	High PE + BC	1311 (39) a	1722 (63) a	4.75 (0.08) ab

Different lowercase letters in the same column indicate significant difference among the treatments (*p* < 0.05).

## Data Availability

Data are contained within the article.

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
