# Peer review of "Biochar Mitigates the Negative Effects of Microplastics on Sugarcane Growth by Altering Soil Nutrients and Microbial Community Structure and Function"

_plants, 2023, doi:10.3390/plants13010083_

Round 1

Reviewer 1 Report

Comments and Suggestions for Authors

Dear Authors,

Yours manuscript is interesting because it raises the important problem of microplastics in the soil environment. In my opinion, the manuscript is well prepared in terms of content. Unfortunately, I noticed some omissions and inaccuracies. Below is a list of comments and observations, both positive and negative:

1.      Please correct the chapter numbering. The numbering should start with the introduction. Pay attention to the chapter titles. What does " 2. Results and Analysis" mean?

2.      Title:I have no comments.

3.      Abstract. This is well done, but the acronyms need to be clarified.

4.      Introduction or discussion. This is well done.

5.      The purpose of the study seems to be well formulated. Unfortunately, as goal 1, you specify the assessment of sugarcane growth under the influence of microplastics and biochar. There is no information on this subject in the latter part of the work. Only in the conclusion do you return to this topic. Please clarify this.

6.      Material and Methods: The methodology is well described. However, the description of several studies is missing: observed_OTUs, Chao1 and Shannon indices. There is also no reference to biomass.

7.      Results. Figure 1.  There is no information on what the lowercase letters in the results mean.

8.      Discussion and Conclusions. These chapters are well written. But explain as in point 5.

9.      Please check the entire text. Lower underlining often appears. What for? Please delete it.

10.  The language appears to be correct, but I don't feel qualified to judge about the English language and style.

Good luck!

Sincerely yours

Reviewer

Author Response

1) Please correct the chapter numbering. The numbering should start with the introduction. Pay attention to the chapter titles. What does " 2. Results and Analysis" mean?

Answer: We have corrected the chapter numbering and started with the introduction. We have check the chapter titles. And “Results and Analysis" have been corrected to “Results”.

2) Title:I have no comments.

Answer: Thanks.

3) Abstract. This is well done, but the acronyms need to be clarified.

Answer: We have clarified the acronyms of operational taxonomic units (OTUs).

4)  Introduction or discussion. This is well done.

Answer: We have revised some text and tried to make the introduction and discussion more clear and concise.

5) The purpose of the study seems to be well formulated. Unfortunately, as goal 1, you specify the assessment of sugarcane growth under the influence of microplastics and biochar. There is no information on this subject in the latter part of the work. Only in the conclusion do you return to this topic. Please clarify this.

Answer: In this study, we have tried to explore the effects of microplastics combined with or without biochar on sugarcane biomass and soil biochemical properties. The goal 1 in the original manuscript is a wrong description and we have revised it to “identify the differences in the biomass of sugarcane, soil physio-chemical properties and microbial community characteristics under different treatmetns of PE plastics combined with or without corn straw biochar”. And we also revised the related content in the parts of Discussion and Conclusions.

6) Material and Methods: The methodology is well described. However, the description of several studies is missing: observed_OTUs, Chao1 and Shannon indices. There is also no reference to biomass.

Answer: Thanks for your suggestions. We have added the calculation process about observed_OTUs, Chao1 and Shannon indices. The part of “Data Analysis” was further supplemented. The components of above-ground and below-ground biomass of sugarcane have been clarified and we also added the reference to biomass.

7) Results. Figure 1.  There is no information on what the lowercase letters in the results mean.

Answer: We have added the information on the lowercase letters in Figure 1.

8) Discussion and Conclusions. These chapters are well written. But explain as in point 5.

Answer: This study aims to explore the effects of microplastics combined with or without biochar on sugarcane biomass and soil biochemical properties in this study. We have revised some content in the Discussion and Conclusions and tried to make our entire work focus on the purpose.

9)  Please check the entire text. Lower underlining often appears. What for? Please delete it.

Answer: The lower underlining is added automatically by the word document. We have delete it in the text.

10) The language appears to be correct, but I don't feel qualified to judge about the English language and style.

Answer: We have employed an English-language editing service to polish our wording.

Reviewer 2 Report

Comments and Suggestions for Authors

After reviewing the manuscript entitled “Biochar Mitigates the Negative Effects of Microplastics on Sugarcane Growth by Changing Soil Nutrients and Microbial Community Structure and Function” I have identified some issues that is required to be addressed. My comments are as follows:

1.       As mentioned in abstract “by a potted experiment” its wrong, write it as “through a pot experiment”

2.       In the abstract mention the increase or decrease of a parameter with supporting data, like the N,P contents decreased by ……%. Do these for all the parameters throughout the manuscript.

3.       The introduction is too long, I will recommend shortening the introduction.

4.       Add reference to the following : Our investigation in the sugarcane area 30 years,……………………………………………………..pollution of microplastics in sugarcane fields is serious.”

5.       In the last paragraph of the introduction do not mention the treatment details as already it is mentioned in the materials and methods section.

6.       Mention the size of the pot as well as the weight of soil used for growing sugarcane.

7.       Clarify what is the meaning of above ground and below ground biomass. Below ground is root hopefully but above ground includes cane, leaves etc. So, clarify it.

8.       As the authors have done redundancy analysis, which is fine, but first they should have done a sample adequacy test. So, I will recommend doing the test and mentioning the results.

9.       I am not convinced behind the reason of decrease of soil pH, N and P content due to the accumulation of polyethylene microplastics. Make the discussion more inclusive, citing some previous work is not sufficient.

Comments on the Quality of English Language

Minor corrections required

Author Response

1)  As mentioned in abstract “by a potted experiment” its wrong, write it as “through a pot experiment”

Answer: We have revised it.

2)  In the abstract mention the increase or decrease of a parameter with supporting data, like the N,P contents decreased by ……%. Do these for all the parameters throughout the manuscript.

Answer: Thanks for your suggestions. We have added these information in the revised manuscript.

3)  The introduction is too long, I will recommend shortening the introduction.

Answer: We have revised it and tried to make it more concise and refined.

4)  Add reference to the following : Our investigation in the sugarcane area 30 years,……………………………………………………..pollution of microplastics in sugarcane fields is serious.”

Answer: Sugarcane mulching has been applied in China for more than 30 years. And our investigation in the sugarcane area showed that the amount of plastic film used in sugarcane fields was 75-150 kg/ha/year. Thus we calculated the possible content of microplastics in the sugarcane fields, it is just a estimated value.

 5)  In the last paragraph of the introduction do not mention the treatment details as already it is mentioned in the materials and methods section.

Answer: We have revised it.

 6)  Mention the size of the pot as well as the weight of soil used for growing sugarcane.

Answer: The pot is 30 cm high, with a top diameter of 32 cm and a bottom diameter of 26 cm. We have added this in the revised manuscript.

7)  Clarify what is the meaning of above ground and below ground biomass. Below ground is root hopefully but above ground includes cane, leaves etc. So, clarify it.

Answer: The above ground includes cane and leaves. We have added this information.

8)  As the authors have done redundancy analysis, which is fine, but first they should have done a sample adequacy test. So, I will recommend doing the test and mentioning the results.

 Answer: We have done it and added this information in the part of “Data Analysis”.

9)  I am not convinced behind the reason of decrease of soil pH, N and P content due to the accumulation of polyethylene microplastics. Make the discussion more inclusive, citing some previous work is not sufficient.

Answer: We have revised this and tried to make it more inclusive. MPs like high-density PE has been proven to decrease soil pH through the cation/proton exchange (Boots et al., 2019). Microplastic have a high adhesion, which can change plant root exudates and soil buffering capacity to co-mediate soil pH (de Souza Machado et al., 2019). Microplastic addition considerably changed microbial community and activities, which may be an important reason for the alteration in soil N and P content (Lian et al., 2021).

Round 2

Reviewer 1 Report

Comments and Suggestions for Authors

I accept the amendments introduced.